# A Review of Recent Studies Employing Hyperspectral Imaging for the Determination of Food Adulteration

**Havva Tümay Temiz [1],\* and Berdan Ulaş [2]**

1   Department of Research and Technology Transfer, Pladis Turkey R&D Center, Kocaeli 41400, Turkey
2   Department of Chemical Engineering, Faculty of Engineering, Van Yuzuncu Yıl University, Van 65080, Turkey; berdanulas@yyu.edu.tr
\*   Correspondence: tumay.temiz@pladisglobal.com

**Abstract:** Applications of hyperspectral imaging (HSI) methods in food adulteration detection have been surveyed in this study. Subsequent to the research on existing literature, studies were evaluated based on different food categories. Tea, coffee, and cocoa; nuts and seeds; herbs and spices; honey and oil; milk and milk products; meat and meat products; cereal and cereal products; and fish and fishery products are the eight different categories investigated within the context of the present study. A summary of studies on these topics was made, and articles reported in 2019 and 2020 were explained in detail. Research objectives, data acquisition systems, and algorithms for data analysis have been introduced briefly with a particular focus on feature wavelength selection methods. In light of the information extracted from the related literature, methods and alternative approaches to increasing the success of HSI based methods are presented. Furthermore, challenges and future perspectives are discussed.

**Keywords:** hyperspectral imaging; feature wavelengths; adulteration; chemometrics; neural networks; wavelength selection

## 1. Introduction

There is a growing need for alternative analytical techniques in the search for rapid, accurate, and reliable quality control systems for large scale production to prevent fraudulent practices and hence to ensure food safety. Food products subjected to adulteration most often are reported to be coffee, tea, juice, wine, spices, olive oil, honey, milk, cereals, meat, fish, and organic food by the European Parliament. In the detailed study of Ulberth et al., extensive use of spectrometry/spectroscopy-based methods to detect adulteration was determined. A lack of internationally accepted validation protocols and the requirement for robust spectral databases of authentic products have been found responsible for the inapplicability of these techniques to routine analyses at present [1]. Within this context, there is an increasing interest in the combined use of optical imaging and spectroscopic techniques for food safety and quality analysis since they provide non-destructive detection, chemical information, and visualization at the same time [2]. Hyperspectral imaging (HSI) is a promising technology that allows simultaneous measurement of spectral and spatial information in a fast and reliable way. The application of chemometric methods is required to extract spectral, textural, and morphological features from high dimensional HSI data [3]. Chemometrics, with its powerful synergy with analytical techniques, provides extraction of essential information buried in high dimensional data sets and helps to reduce the abstrusity of the data [4].

Simultaneous analysis of the samples is one of the unique advantages of HSI based techniques that are not possible for other spectroscopic techniques [5]. NIR hyperspectral imaging (NIR-HSI), Raman hyperspectral imaging (Raman-HSI), and hyperspectral fluorescence imaging (HSFI) are three HSI techniques frequently used for the determination of food quality and safety. The fact that it collects a high amount of information and

provides high accuracy discrimination has enabled the NIR-HSI technique to find a wide variety of applications in food analysis [3,6–9]. Visible (380 to 800 nm), ultraviolet (200 to 400 nm), VIS/NIR (400 to 1000 nm), NIR (900 to 1700 nm), and short wave infrared (970 to 2500 nm) are the spectral regions that are widely used for HSI based food analysis. There are a number of difficulties reported for IR imaging, which are water absorption of the IR radiation, low spatial resolution due to long IR wavelengths, and low signal intensity due to background interference [10].

Through the combination of Raman spectroscopy and imaging, Raman-HSI or Raman chemical imaging allows researchers to visualize chemical properties and distribution of components by acquiring spectral and spatial information at once [11]. Since it is possible to perform measurements at the micro or nano-scale level, limitation of HSI to macro-scale level could be overcome by Raman-HSI [8]. Allowing label-free detection and simultaneous imaging of the spatial distribution of multiple chemical species through their fingerprint Raman spectra, which is comprised of well-resolved peaks, is an additional advantage of Raman-HSI. The very weak Raman signal of water makes Raman-HSI suitable for the analysis of aqueous solutions. Several review articles are summarizing the principles and applications of Raman-HSI for food quality and safety evaluation [2,12]. To increase data acquisition rates, eliminate fluorescence backgrounds, and enhance the quality of Raman signals, research on Raman based imaging methods is essential. Coherent anti-Stokes Raman scattering (CARS) imaging, stimulated Raman scattering (SRS) imaging, surface-enhanced Raman scattering (SERS) imaging, and tip-enhanced Raman scattering (TERS) imaging are some of the recently developed techniques [2]. Raman HSI is a relatively new technique compared to NIR HSI methods. The promising potential of this technique for food quality and authenticity studies was emphasized in [13], but there are still limited studies employing this technique for food analysis.

Most of the HSI based studies conducted on food and agricultural products have employed hyperspectral reflectance imaging technology, which measures reflectance in the visible region to short-wave infrared region. Fluorescence phenomenon is described as the absorption of short-wavelength light by the molecules and its subsequent emission as light with a longer wavelength, and this phenomenon provides compositional information about the analyzed sample at the same time. Hence, like reflectance, it is possible to measure fluorescence both spectroscopically or by HSI [14]. The low rate of data acquisition seems to be one of the main barriers to HSFI being used in real-time applications. However, efforts are being made to develop HSFI systems using high-speed computers combined with fast data analysis algorithms. Not having fluorescent properties of all components of food is the inherent disadvantage of the technique. Miniaturization of the system components and development of portable systems are described as the near trends of HSFI [15]. Emphasis has been put on the use of alternative light sources, which quickly induce fluorescence while not generating too much heat. The operation of HSFI in transmittance mode and combined use with reflectance-HSI are other subjects that have been pointed out. Applications of HSFI for quality and safety analyses of food and agricultural products have been summarized in several articles [16,17].

There are several review articles on the use of HSI for food analyses. Studies about the use of HSI on different food categories, namely fruit, vegetables, meat, seafood, and grains, have been gathered by Huang et al. [5]. The use of HSI for quality and safety determination of fruits and vegetables has been summarized in [9]. The main components of HSI were represented, and applications of the multispectral imaging (MSI) and HSI for safety and quality evaluation of fruits and nuts, vegetables, meat, grains, and beverages were reported [18]. In other review articles, studies on food quality and safety control were classified based on the data acquisition mode; reflectance, fluorescence, transmittance, interactance, and scattering [3,19]. Unlike the abovementioned articles, this study has explicitly focused on the detection of food adulteration in different categories. These are tea, coffee, and cocoa; nuts and seeds; honey and oil; herbs and spices; milk and milk products; meat and meat products; cereal and cereal products; and fish and fishery

products. Feature wavelengths determined by different algorithms in recent reports were tabulated based on food categories included in this article. Ways to increase success and alternative approaches are offered in HSI based methods. Moreover, challenges and future perspectives are discussed.

## 2. Applications of HSI to Detect Food Adulteration

### 2.1. Tea, Coffee, and Cocoa

Hyperspectral image data collected between 408 to 1117 nm was analyzed using the support vector machine (SVM) as a pattern classifier to differentiate five grades of roasted green tea. Principal component analysis (PCA) was used to reduce the dimension of the data and to select optimal band images. The performance of artificial neural networks (ANN) was compared to those of *linear discriminant analysis* (LDA), back-propagation ANN (BP-ANN), and radial basic function ANN (RBF-ANN) [20]. Vis-HSI, coupled with maximum likelihood classifier (MLC) and ANN was used to classify five different Chinese tea samples based on data between 400 and 800 nm. PCA was applied to extract uncorrelated components from highly correlated data [21]. Hyperspectral images captured in the range of 920 to 2514 nm were used to differentiate herbal tea raw materials and quantify the composition of herbal tea blends. PCA and partial least squares-discriminant analysis (PLS-DA) models were developed [22]. In another study conducted on quality assessment of herbal tea blends, hyperspectral images acquired between 1000 to 2500 nm were analyzed using PCA and PLS-DA [23]. Sixteen green tea samples from seven different geographical origins were differentiated using a fusion of textural and NIR-HSI data at raw data-, feature- and decision-levels where the highest classification efficiency was obtained through raw data-level fusion. The spectral range used in this study was reduced to 967 to 1700 nm by the application of the Savitzky-Golay filter to remove noise. Error-correcting output code (ECOC)-SVM models were developed to differentiate the samples [24].

NIR-HSI was employed, and data between 955 and 1700 nm was collected to classify Arabica and Robusta coffee species. Sparse methods, namely sparse PCA (sPCA) + kNN and sparse PLS-DA (sPLS-DA), were compared with the classical methods, namely PCA + kNN and PLS-DA [25]. Hyperspectral images captured between 874 and 1734 nm were used to identify four different Chinese coffee bean varieties. Performances of prediction maps and SVM discrimination models constructed using full sample average spectra, pixel-wise spectra, and the selected optimal wavelengths by second derivative spectra were compared [26]. Roasting and grinding processes enhanced the ease of adulteration in coffee and made it one of the most tempting targets of fraudulent practices in the food industry. Roasted and ground coffee samples adulterated with coffee husks, roasted and powdered corn kernels, wood sticks, and soil were analyzed using FT-NIR-HSI and the multivariate curve resolution with alternating least squares (MCR-ALS). Pure Robusta coffee bean certified by producers was used as the reference sample, and its spectrum was compared to the MCR recovered spectra to identify the adulterant. Collected data was between 4000 and 7800 cm$^{-1}$. Additionally, a control chart was created, which is composed of MCR-ALS scores of pure samples and adulterants at varying ratios. Warnings and action limits were defined in this control chart to determine whether the unknown sample is adulterated or not [27].

There is increasing consumer demand for the verification and certification of cocoa beans imported from the different geographical origin of production. Fermented and dried Forastero cocoa beans from Africa, America, and Southeast Asia were classified by proton transfer reaction-quadrupole interface-time of flight-mass spectrometry (PTR-QiToFMS) and HSI. Volatile profiles and spectra were used to create a fingerprint for each cocoa sample. Pearson correlation test was used to correlate the results of two different techniques, but only an indirect relationship was found. A high degree of separation between the African and American samples and high variability within Southeast Asian samples were reported [28].

## 2.2. Nuts and Seeds

Jiang et al. have analyzed the potential of fluorescence hyperspectral imaging (FHSI) coupled with several data analysis methods, namely the Gaussian-kernel based SVM approach [29] and PCA-Gaussian mixture model (GMM) based method [30] to differentiate shell and meat parts of walnut samples. Independent component analysis with k nearest neighbor classifier (ICA-kNN) was used for the analysis of the data between 425 and 775 nm [31]. Raman spectral imaging coupled with PCA and PLSR was used to quantify adulteration of pistachio nut granules with green pea granules. Raman spectral data was between 200 and 3700 cm$^{-1}$ [32].

HSI was used to discriminate the basil seeds based on their country of origin and to predict their physicochemical properties, namely moisture content, crude lipid content, total phenolic compounds content, fatty acid content, and color. Collected data between 900 and 1700 nm was analyzed using PCA and PLS-DA [33].

## 2.3. Honey and Oil

The application of HSI for the analysis of liquid and semi-liquid foods is less common. Baiano et al. summarized the studies on classification, determination of adulteration, and evaluation of quality parameters of edible oils like virgin olive oil, sesame oil, and frying oils [34]. Hyperspectral images were obtained for olive oil samples to evaluate their correlation with reference methods in terms of free acidity, peroxide index, and moisture contents. Genetic algorithm (GA), least absolute shrinkage, and selection operator (LASSO), and successive projection algorithm (SPA) were chosen to determine the optimal wavelengths [35]. Biochemical changes that occur during the ripening process of olive fruit were monitored using FTIR imaging coupled with PCA and PLS-DA. Olive oil accumulation in flesh and modifications of cell wall polysaccharides in flesh and hypodermis were determined. Temporal changes in flesh and hypodermis tissues were monitored by evaluating the spectral intensity changes in PLS loading vectors [36]. Quality characteristics of sesame oils produced from different raw materials, namely whole-sesame seeds, sesame powder, and the mixture of both, were determined using HSI. PLS-DA and PCA were used for data analysis [37].

The potential of HSI to detect adulterated honey samples by the addition of sugar syrup has been evaluated. VIS-NIR images were analyzed using ANN, SVM, LDA, Fisher, and Parzen classification methods [38]. The VIS-NIR HSI system was used to discriminate honey samples of acacia, lime, buckwheat, rapeseed, and heather. Machine learning techniques, namely, radial basis function (RBF) network, SVM, and random forest (RF), were employed [7]. Segmentation and calibration techniques were purposed to eliminate distortions due to temperature and lighting fluctuations and to obtain reproducible spectra of honey samples using HSI in reflectance and transmittance modes [39]. Reflectance spectra obtained by a VIS-NIR HSI system that collects data between 399.40 to 1063.79 nm were evaluated to classify honey samples based on their botanical origin. In the related study, a new classification algorithm consisting of two successive stages was used. 'One class classification' and 'main class classification' allows the system to filter out invalid inputs and allow only familiar inputs to proceed to the second stage [40].

Feature wavelengths reported to be utilized for discrimination or determination of adulteration in tea, coffee, cocoa, nuts, seeds, honey, and oil samples were summarized in Table 1. Molecules responsible for related wavelengths chosen as the discriminant wavelength were indicated in most of the studies. Collected data was frequently in a similar wavelength range. Although a wide variety of algorithms were employed to detect these wavelengths, the loadings plot of PCA and PLS-DA are still the most commonly used wavelength selection methods.

**Table 1.** Feature wavelengths reported for tea, coffee, cocoa, nuts, seeds, honey and oil.

| Type of Adulteration | Spectral Range (nm) | Selected Wavelengths (nm) | Wavelength Selection Method | Assignment of the Wavelengths | Reference |
|---|---|---|---|---|---|
| Different roasting levels in green tea | 408–1117 | 762, 793, 838 | Local maximum weighing coefficients of PCA | No assignment | [20] |
| Two different *Sceletium* species in herbal tea blends | 1000–2500 | 1874–2061, 2061–2248, 1436, 2123 | High weighted loadings of PCA | No assignment | [23] |
| Green tea from different geographical origin | 967–1700 | 1381 | PSNR and SSIM | No assignment | [24] |
| Classification of cocoa beans from different geographical origins | 400–1000 | 600–700 700–730, 870–910 770–830 | Loading values of PCA | Color Organ, compounds, Fatty acids Aminoacids | [28] |
| Identification of coffee bean varieties | 874–1734 | 995, 1005, 1019 1129, 1139, 1210, 1214, 1241 1342, 1372, 1399 1409, 1440, 1443, 1460 1483 1500 1507, 1534 1609 1629 | Peaks and valleys with large differences in second derivative spectra | 2nd overtone of N-H stretch 2nd overtone of C-H stretch C-H vibrations Water 2nd overtone of O-H stretch CH$_2$ stretching and nonstretching 1st overtone of N-H stretch 1st overtone of C-H stretch Aromatic C-H band | [26] |
| Classification of Arabica and Robusta coffee species | 955–1700 | 1143 1446 1410 1420 1195–1225 | Loading vectors of PCA, PLS-DA, sPCA and sPLS-DA | C-H aromatic 2nd overtone C-H combination band O-H 1st overtone of aliphatic alcohol O-H 1st overtone of aromatic alcohol C-H aliphatic 2nd overtone | [25] |
| Walnut shell and walnut meat differentiation | 425–775 | 456.5, 443, 429.5, 447.5, 438.5 | ICA based optimal band selection approach | No assignment | [31] |
| Discriminating the origin of *Ocimum basilicum* L. | 900–1700 | 1450–1457, 1242–1254, 1380, 1696 | Beta coefficients of PLS-DA | Moisture, Lipid, Phenolic contents and Fatty acids | [33] |
| Green pea adulteration in pistachio nut granules | 200–3700 cm$^{-1}$ | 1441, 1655 cm$^{-1}$ | Changes in band intensity | Lipid content | [32] |
| Discriminating the floral origin of honey | 400–1000 | 425 | Loadings plot | Color | [7] |
| Quality parameters of olive oil | 900–1700 | 1000–1160, 1280–1350, 1480–1500, 1570–1640 1210–1240, 1390–1430, 1630–1670 1206–1241, 1390–1440, 1447–1594, 1640–1660 | GA SPA SPA | Free acidity Peroxide index Moisture content | [35] |
| Discrimination of olive fruits at different stages of maturation | 750–4000 cm$^{-1}$ | 900–1200 1072 1034 1101 1513, 1606, 1626 1175, 1462, 1747, 1750 1395, 1582 | Loading vectors of PLS-DA | Pectic polysaccharides and hemicellulose Galactose Glucose Pectin Phenols Olive oil triglycerides Depolymerisation and de-esterification of cell wall polymers | [36] |
| Discrimination of sesame oils | 900–1700 | 1149, 1442, 1673, 1693 | Beta coefficient of PLSDA model | Fatty acid composition | [37] |

PCA: principal component analysis, PLS-DA: partial least squares-discriminant analysis, PSNR: peak signal-to-noise ratio, SSIM: structural similarity index measure, SPA: successive projection algorithm, GA: Genetic Algorithm, ICA: independent component analysis, sPCA: sparse-principal component analysis, sPLS-DA: sparse-partial least squares-discriminant analysis.

### 2.4. Herbs and Spices

Herbs and spices are often exposed to fraudulent practices through the addition of cheaper materials that resemble the color or appearance of the main ingredient. The use of visual and microscopic inspection methods to detect these types of food adulterations has been recommended by the British Retail Consortium, the Food and Drink Federation, and the Association of Spices in the Guide on the Authenticity of Herbs and Spices [41].

HSI imaging was used to distinguish between two similar species of *Sceletium tortuosum* and *Sceletium crassicaule* since *S. tortuosum* has gained massive interest due to its potential use in the treatment of several psychological conditions. Images were analyzed using PCA and PLS-DA [42]. Potential of HSI to determine the substitution or adulteration of *Ilicium verum* (Chinese star anise) with *Illicium anisatum* (Japanese star anise) was investigated since Japanese star anise is known to be highly toxic. Hyperspectral image data was analyzed using PCA and PLS-DA [43]. HSI was employed to determine incorrectly labeled commercial products and raw materials of *Echinacea* species. Authentic species, namely, *Echinacea angustifolia*, *Echinacea pallida*, and *Echinacea purpurea*, are included in various treatment formulations for upper respiratory tract infections. A clear differentiation was obtained through the analysis of collected data by PCA and PLS-DA [44]. HSI, coupled with PCA and PLS-DA, was used to differentiate *Stephania tetrandra* and *Aristolochia fangchi* root powder while the latter is known to contain aristolochic acid that causes urothelial carcinoma and aristolochic acid nephropathy [45].

The presence of millet and buckwheat flours in black pepper samples was determined using a NIR-HSI system. PCA and PLS-DA were the methods of choice for data analyses [46]. Quantification of papaya seed, as one of the most widely used adulterants of black pepper, in black pepper powder and black pepper berry samples was investigated using the NIR-HSI system. PCA, soft independent modeling of class analogy (SIMCA), and PLSR were used as the data analysis methods [47]. The handheld HSI system was employed to authenticate nutmeg samples and to predict the adulteration ratio. PCA, PLS-DA, and artificial neural networks based on multilayer perceptron (ANN-MLP) were used for the analysis of collected data [48]. *Zanthoxylum bungeanum* is a type of condiment that has found applications in medical and food industries due to its aroma, numbing spiciness, and therapeutic properties, which are influenced by its geographical origin. HSI, coupled with SVM, was used to evaluate the data obtained between 380 and 1040 nm to classify the samples based on their geographic origin [49]. An MSI system was employed to determine turmeric powder samples adulterated with tartrazine colored rice flour. A relationship was established between the adulteration ratio and Bhattacharyya distance through a second-order polynomial. Multispectral images were captured using LEDs with nine different peak wavelengths, namely, 405 nm, 430 nm, 505 nm, 590 nm, 660 nm, 740 nm, 850 nm, 890 nm, and 950 nm. It is emphasized in the study that a small number of spectral bands and low resolution of captured images diverted the researchers to develop a five-stage data analysis algorithm consisting of (i) dark current subtraction, (ii) adaptive Weiner filtering, (iii) PCA based dimension reduction, (iv) multivariate Gaussian model construction, and (v) construction of the functional relationship between the Bhattacharyya distance and the adulteration levels [50].

Table 2 shows the feature wavelengths utilized for discrimination or determination of adulteration in herbs and spices. A lack of assignment for feature wavelengths was determined for most of the samples in these categories.

**Table 2.** Feature wavelengths reported for herbs and spices.

| Type of Adulteration | Spectral Range (nm) | Selected Wavelengths (nm) | Wavelength Selection Method | Assignment of the Wavelengths | Reference |
|---|---|---|---|---|---|
| Three *Echinacea* species in commercial products | 920–2514 | 1937–2400 | Loadings line plots of the first vector in PCA | No assignment | [44] |
| *Aristolochia fangchi* in root powders of *Stephania tetrandra* | 920–2514 | 964–1474, 1323–1434 964–1322,1435–1474 | Trend Tool PCA Loadings plot PCA Loadings plot | No assignment *Aristolochia fangchi* *Stephania tetrandra* | [45] |
| Adulteration of *Ilicium verum* with *Illicium anisatum* | 920–2514 | 1254–1342, 1737–1887, 2049–2179 1504–1530, 1905–1993, 2254–2297 | Loadings line plots | *Illicium anisatum* *Ilicium verum* | [43] |
| Millet and buckwheat flour in black pepper | 1000–2498 | 1461, 1995, 1999, 2241, 2303, 2347 | Loadings line plots | Protein and oil content | [46] |
| Papaya seeds in powders and berries of black pepper | 900–1700 | 1029, 1242, 1385, 1494, 1518, 1584, 1669 | Beta regression coefficients of PLSR | Phenols, flavonoids, quinines, starch in black pepper Fiber, protein, phenols, quinines in papaya seeds | [47] |
| Pericarp, creamy spent, brown spent and shell in nutmeg | 400–1000 | 400–500, 650–850, 950–1000 | Visual inspection | No assignment | [48] |
| Discriminating the origin of *Zanthoxylum bungeanum* | 380–1040 | Not specified. | CARS and VCPA | No assignment | [49] |

PCA: principal component analysis, PLSR: partial least squares regression, CARS: competitive adaptive reweighted sampling, VCPA: variable combination population analysis.

### 2.5. Milk and Milk Products

NIR-HSI was employed to determine the presence and location of melamine particles in nonfat milk powders at low levels. Data were evaluated using spectral similarity analysis methods, namely spectral correlation measure, spectral angle measure, and Euclidian distance measure, which resulted in similar performances [51]. The same wavelength range was utilized in another study by the same research group in which NIR hyperspectral images were converted to reflectance images. Several PLSR models with different preprocessing methods were developed to determine melamine particles in nonfat milk powder [52]. Hyperspectral images were collected using an NIR-HSI system to quantify melamine particles in nonfat and whole milk powders. A linear correlation algorithm was used to determine optimal bands, and then band ratio images were obtained for these two wavelengths. A successful application of NIR-HSI coupled with band ratio methodology for melamine detection was reported [53]. Studies on the use of HSI for turbid liquid and semi-liquid food analyses such as determination of milk adulteration or discrimination of different yogurt microstructures, have been summarized in a review article [34].

HSI was used to determine corn starch levels in adulterated fresh cheese samples. Collected data were analyzed using PLSR. Effects of moisture content in cheese and type of adulterant starch on the success of the model were emphasized in this study [54]. Commercial cheddar cheeses of four different brands were classified using NIR-HSI coupled with PLS-DA, LDA, and SPA-LDA. PLS-DA regression coefficients were used to determine the wavelengths differentiating cheddar cheeses of different brands. Higher PLS-DA model performances were obtained by utilizing HSI data rather than texture and color data [55].

Table 3 shows the feature wavelengths utilized for discrimination or determination of adulteration in milk and milk products. The determination of melamine adulteration has

been a hot topic in this category. Although almost the same wavelength interval was used in different studies, research was focused on employing different wavelength selection methods to increase accuracy.

**Table 3.** Feature wavelengths reported for milk and milk products.

| Type of Adulteration | Spectral Range (nm) | Selected Wavelengths (nm) | Selection Method | Assignment of the Wavelengths | Reference |
|---|---|---|---|---|---|
| Melamine in milk powder | 990–1700 | 1473.8 | Spectral similarity analysis | Melamine content | [51] |
| Melamine in milk powder | 990–1700 | 1478.6, 1468.9 | Beta coefficients of PLSR | Melamine content | [52] |
| Melamine in milk powder | 938–1654 | 1447, 1466 | Band ratio algorithm | Melamine content | [53] |
| Starch in fresh cheese | 200–1000 | 928, 984 | Beta coefficients of PLSR | Water content | [54] |
| Classification of commercial Cheddar cheeses from different brands | 950.35–1654.15 | 1190<br>1124<br>1268, 1271<br>1367, 1370<br>1235<br>1055, 1171<br>1364<br>1152, 1312 | Regression coefficients of PLSR | Fat content<br>L* value<br>a* value<br>a* value<br>Fat content, Enzyme treatment<br>Protein and total saturated fatty acids<br>Protein content, pH value<br>L* value, Water holding capacity | [55] |

PCA: principal component analysis, PLSR: partial least squares regression.

### 2.6. Meat and Meat Products

The European horse meat scandal in 2013 has been a warning sign for food safety authorities, food scientists, and consumers to bear in mind how vulnerable the food supply chain is [56]. In a very comprehensive report on the global beef supply chain, primary and secondary processing and farming were determined as the most vulnerable steps in which counterfeiting and adulteration were the most frequent type of fraud [57].

In the early studies, MSI analysis coupled with neural networks was employed to separate wholesome carcasses from unwholesome carcasses during an on-line poultry carcass inspection [58]. Elmasry et al., has published a detailed review article discussing the promising potential of HSI and other imaging techniques compared to traditional methods for determination of meat quality in terms of color, quality grade, marbling, maturity, texture, and simultaneous measurement of multiple chemical constituents [59]. Several authors investigated the differentiation of fresh and frozen-thawed meat using HSI coupled with texture analyses. Hyperspectral reflectance images were analyzed using PLS-DA and probabilistic neural networks (PNN). Their results were given with a correct classification ratio, sensitivity, specificity, and accuracy values, as well as confusion matrices, receiver operating characteristic (ROC) analysis, and the value of the area under the ROC curve (AUC) [60–62].

Several studies were reported on the use of HSI for the determination of meat adulteration. In these studies, partial least squares regression (PLSR) models were constructed to detect the adulteration ratio. Although similar results were obtained using full spectra, the importance of the selection of the specific wavelengths was emphasized since it reduces the dimension of hyperspectral data [63–65]. PLS-DA models discriminated pork, poultry, and fish protein meals with high classification ratios through the integration of spectral and textural information extracted from NIR-HSI data [66]. Vis/NIR-HSI was employed to predict the fresh meat adulteration with spoiled meat of the same origin. PLSR, SVM,

least squares support vector machine (LS-SVM), and extreme learning machine (ELM) were employed to develop models with adequate coefficients of determination ($R^2$) and RMSEs [67]. Duck meat adulteration in beef was determined using VIS-NIR HSI coupled with PLSR and principal component regression (PCR). PCA loadings and two-dimensional correlation spectroscopy (2D-COS) were used to select feature wavelengths. The optimal wavelength selection method of PC resulted in better prediction performance. Adulteration distribution maps were generated through the transfer of the PLSR model to each pixel in the image [68]. Due to their high heterogeneity, meat and meat products have been frequently analyzed using HSI techniques [68]. A method based on a visual appraisal to determine minced beef adulteration with minced chicken was developed using VIS-NIR HSI operating between 380 and 1000 nm. Gaussian distribution of regression coefficients (GD-RC) model was used for data analysis, and its performance was compared to that of SVM, kNN, and decision tree (DT) methods. The ratio of chicken pixels to total meat pixels was described as the adulteration ratio [69]. VIS-NIR HSI and machine learning algorithms were used to quantify minced beef and minced pork adulteration with chicken and textured vegetable protein. Hyperspectral data between 400 and 1000 nm was utilized to construct classification models by feed-forward artificial neural networks (FFNN), decision trees, kNN, LDA, PLS-DA, and SVM. PLSR was used to determine the adulteration ratio. Sequential forward selection (SFS) and interval partial least square (IPLS) were used for wavelength selection of classification and adulteration models. However, both algorithms have come out with too many feature wavelengths and this result has been associated with the large number of adulterant levels investigated in the study. It was emphasized in the study that using selected wavelengths does not always guarantee higher model performance [70]. Minced pork adulterated with minced pork jowl meat was determined using PLSR coupled with NIR-HSI acquiring data between 400 and 1000 nm. The jowl is not convenient for human consumption since it is described as a kind of lymphatic meat with a significant amount of lymph nodes. Three different wavelength selection methods, namely regression coefficients of PLSR, wavelengths selected by two-dimensional correlation spectroscopy, and loadings of PCA, were employed to eliminate useless wavelengths. Selected wavelengths by each method have individually been used to construct PLSR models. The best results were obtained when regression coefficients were utilized while the performance was slightly decreased compared to full spectra models [71]. Selected wavelengths and their molecular assignments were summarized in Table 4 for only very recent articles. A comparison was made between line scanning HSI and NIR and VIS snapshot HSI in terms of their efficiency to classify red meat products, namely lamb, beef, and pork. A deep 3D convolution neural network (3D-CNN) model has been employed to extract spectral and spatial features out of collected data. In order to improve the performance of 3D-CNN, a graph-based post-processing method was also developed. Although accuracy was comparatively low for snapshot HSI, a new subject has been set forth for further research [72]. Specific wavelengths for predicting meat quality attributes, detecting safety parameters in meat, and authentication of meat were summarized [73]. Components of HSI systems, operating spectral ranges, and data analysis methods used for safety and quality detection of chicken meat were documented. Urgent need for cameras with higher spectral and spatial resolution, alternative light sources and measures to be taken to reduce the specular reflection, and importance of the developments in artificial intelligence algorithms were indicated [74].

**Table 4.** Feature wavelengths reported for meat and meat products.

| Type of Adulteration | Spectral Range (nm) | Selected Wavelengths (nm) | Selection Method | Assignment of the Wavelengths | Reference |
|---|---|---|---|---|---|
| Minced pork adulteration with minced pork jowl meat | 400–1000 | 440<br>491<br>545, 560, 570, 752<br>632<br>686<br>871<br>491, 632, 871<br>433, 450, 481, 558, 578, 594, 634, 661, 889, 948 | Loading lines of PCA<br>2D-COS<br>Regression coefficients of PLSR | Deoxymyoglobin<br>Metmyoglobin<br>No assignment<br>Sulfmyoglobin<br>Redness<br>Hydrocarbons<br>No assignment<br>No assignment | [71] |
| Beef adulteration with duck meat | 400–1000 | 605, 676<br>948<br>505, 537, 576, 605, 636, 676, 948 | 2D-COS<br>Loading lines of PCA | Red color<br>Water content<br>No assignment | [68] |

PCA: principal component analysis, PLSR: partial least squares regression, 2D-COS: two-dimensional correlation spectroscopy.

### 2.7. Cereal and Cereal Products

Detection and quantification of entire or broken ergot bodies along with other contaminants such as rapeseed and straw pieces in cereals, namely rye, organic rye, organic triticale, oats, black oats, and barley, was performed using an NIR-HSI system. Transfer of the developed protocol from laboratory level to industrial level was completed successfully within the context of this study [75]. NIR-HSI was employed to discriminate between wheat and three different types of contaminants. The naive Bayes (NB), SVM, and k-nearest neighbors (k-NN) classifiers were used to evaluate the obtained data [76]. The NIR-HSI system was employed to discriminate between oat, barley, wheat, and rye. PCA and PLS-DA classification models were developed for this purpose. The application of a wavelength selection algorithm called variable importance in projection (VIP) has provided better prediction efficiency [77]. Detection of durum wheat contamination with common wheat was performed based on four different approaches, namely morphological criteria, NIR spectral profile, protein content criteria, and vitreousness, using PLS-DA coupled with NIR-HSI [78]. Five different added fibers in various ratios and their distribution in three different semolina samples were determined by PCA, SIMCA, and PLSR models constructed using NIR-HSI data [79]. Spectral, textural, and morphological features extracted from HSI data coupled with principal component analysis network (PCANet) was used to determine rice adulteration. Accuracy levels obtained using PCANet were compared with those of KNN and random forest [80].

Peanut adulteration in wheat flour was quantified using NIR-HSI operating between 1000 and 2200 nm. Collected data were analyzed using PCA [81]. In the subsequent research of the same group, the potential of NIR-HSI, coupled with independent components analysis (ICA), employing a joint approximation diagonalization of eigen-matrices (JADE) algorithm to determine peanut adulteration in wheat flour was investigated. Capabilities of PCA and ICA were compared, and the superiority of ICA due to its ability to recover the source signals was emphasized [82]. Peanut powder contamination in spring wheat flour and winter wheat flour was determined using NIR-HSI, while competitive adaptive reweighted sampling (CARS) was used to select the optimal wavelengths. PLSR models were developed for quantitative adulteration analysis [83]. Through the identification of each pixel as an adulterant or sample, the detection of defatted peanut adulteration in wheat flour was investigated at pixel scale using NIR-HSI coupled with a matched subspace detector (MSD) algorithm. Unlike from the previous ones, in the related study, researchers focused on proposing a solution based on the linear mixing model of the subpixel detection

problem that occurs when the particle size of the analyzed samples is smaller than the pixel size [84].

The potential for short wave-NIR-HSI to detect the adulteration of wheat flour and bread with sorghum, oat, and corn was investigated [85]. A new wavelength selection approach called first-derivative and mean centering iteration algorithm (FMCIA) was developed to determine the adulteration of organic spelt (*Triticum spelta* L.) flour with rye flour and organic wheat flour and spelt flour. Afterwards, PLS-DA, MLR, and PLS models were constructed by employing selected optimal wavelengths, and a multispectral real-time imaging system was proposed [86]. In another research reported by the same group, the potential of HSI to detect Irish organic wheat flour adulteration by common wheat flour, cassava flour, and cornflour was investigated. Feature wavelengths were determined based on loading plots of both PCA and FMCIA, while the superiority of the letter method was emphasized [87]. Hyperspectral image data was collected for cooked millet flour, cooked soybean flour, and two adulterated flours. PCA, SPA, and CARS methods were used to select feature wavelengths, while classification models were constructed using LS-SVM [88]. Studies employing HSI for safety and quality issues of cereal and cereal products were summarized in a review article [89].

Binary mixtures of cornstarch and icing sugar with similar particle size and particle density were analyzed using NIR-HSI and Raman HSI systems. PCA and PLSR were used for data analysis. The main aim of the study was to compare the potential of two different techniques for food powder analysis, while slightly better results were obtained with NIR-HSI [90]. Line-scan Raman hyperspectral imaging (RHI) employing a 785 nm laser line was used to visualize and determine multiple wheat flour adulteration with benzoyl peroxide, alloxan monohydrate, and L-cysteine. Pixels belonging either to flour background or adulterant were discriminated using spectral angle mapping (SAM) [91]. NIR-HSI was employed to localize and discriminate talcum powder and benzoyl peroxide in wheat flour both individually and synchronously. The first derivative band difference, spectral correlation measurement method, and band ratio method were used to discriminate pure samples from mixture samples [92]. Simultaneous discrimination and quantification of seed, non-seed, and grain ingredients in multigrain flour mixes were investigated using HSI operating at the visible and near-infrared range. The need for high acquisition speed, high spatial resolution, and high discrimination power was emphasized since an industrial application was planned. The performance of the developed method was compared with regular color imaging, while lower accuracy was obtained for the latter. LDA, the quadratic discriminant classifier (QDC), SVM, random forest (RF), and ANN were used for classification [93].

A wide range of spectral intervals was employed to detect adulteration in cereal and cereal products. They were summarized in Table 5. Loadings plot of PCA, PLS, and PLS-DA is the most commonly used wavelength selection method. Water, protein, and starch contents are the major molecules responsible for the discriminant wavelengths.

## 2.8. Fish and Fishery Products

In terms of food safety and consumer awareness issues, false declaration of geographical origin or production method, mislabeling of frozen-thawed or cold-stored products as fresh, and substitution of high priced products with cheap alternatives are significant problems encountered in the seafood industry. Several researchers have reviewed studies employing HSI for the determination of fish freshness and quality [6,94].

**Table 5.** Feature wavelengths reported for cereal and cereal products.

| Type of Adulteration | Spectral Range (nm) | Selected Wavelengths (nm) | Wavelength Selection Method | Assignment of the Wavelengths | Reference |
|---|---|---|---|---|---|
| Peanut adulteration in wheat flour | 1000–2200 | 1200, 1395, 1734 1450, 1580, 1940, 2100 | Loadings plot of PCA | Higher content of long chain fatty acids Starch and water content | [81] |
| Peanut adulteration in wheat flour | 1000–2200 | 1200, 1395, 1734 1580, 2100 1450, 1940 2030 | Independent Components of ICA | Higher content of long chain fatty acids Starch content Water content Amide content | [82] |
| Peanut adulteration in wheat flour | 935.61–1720.23 | 1196, 1354, 1411, 1478, 1482, 1492, 1545 1200, 1203, 1242, 1245, 1249 | Loadings plot of PLSR | C-H 2nd overtone from $CH_3$, C-H combination band from $CH_3$, O-H 1st overtone from ROH in oil, N-H stretch 1st overtone from $CONH_2$ and CONHR, O-H stretch 1st overtone in starch Protein and starch content | [83] |
| Adulteration of organic spelt flour | 897–1753 | 1145, 1192, 1222, 1349, 1359, 1396, 1541, 1567 | Loading plot of StdDev coefficient resulting from FMCIA | No assignment | [86] |
| Discrimination of oat from barley, wheat, and rye | 900–1700 | 1069, 1126, 1189, 1243, 1413 | VIP | No assignment | [77] |
| Adulteration of cooked millet flour | 865–1711 | 935, 968, 1011, 1117, 1207, 1297, 1416, 1567 1084, 1130, 1207, 1230, 1330, 1426, 1552 1184, 1204, 1323, 1393, 1420, 1479, 1556 | PCA SPA CARS | No assignment | [88] |
| Discrimination of durum wheat from common wheat | 1100–2400 | 1420, 1910;1702,2274;1979, 2054, 2199 1420, 1947; 1677, 2330; 1476, 2023, 2230; 2117 | Loadings plot of PLS-DA model on protein content Loadings plot of PLS-DA model on vitreousness | Water; Fat; Protein and Gluten content Water; Fat; Protein and Gluten; Starch content | [78] |
| Prediction of corn flour content in icing sugar samples | 880–1720 | NIR-HSI 1391, 1419, 1426, 1454, 1482, 1503 Raman-HSI 164.3, 167, 169.7, 172.4, 459.3, 515.6, 551.2, 553.7, 579, 581.5, 599.1 | Ensemble Monte Carlo Variable Selection | No assignment | [90] |
| Identification of fiber added to semolina | 928–2524 | Not specified. | Loadings plot of PCA | Water, Starch and Cellulose content | [79] |

PCA: principal component analysis, PLSR: partial least squares regression, ICA: independent component analysis, SPA: successive projection algorithm, PLS-DA: partial least squares-discriminant analysis, VIP: variable importance in projection, CARS: competitive adaptive reweighted sampling, FMCIA: first derivative and mean centering iteration algorithm.

Discrimination was targeted between fresh, fast frozen-thawed, and slow frozen-thawed fish samples using LS-SVM classifiers applied to VIS-NIR hyperspectral data between 380 and 1030 nm. Data acquisition was performed on the samples equilibrated to room temperature. PCA was used as the data reduction method, while the Gray-level co-occurrence matrix (GLCM) was employed to attain textural variables out of images [95]. The freshness of rainbow trout samples that were kept on ice for 1, 3, 5, and 7 days were

compared using VIS-NIR HSI operating between 400 and 1000 nm and SW-IR HSI working between 1000 and 2500 nm while the latter system has resulted with better prediction efficiency. PCA and PLS-DA were used for data analysis. Savitzky-Golay (SG) function for smoothing, standard normal variate (SNV), multiplicative scatter correction (MSC), first derivative, second derivative, and several combinations of these preprocessies were applied to remove unwanted physical effects in the collected spectra [96]. Fresh, cold-stored, and frozen-thawed samples of shelled shrimp were discriminated using VIS-NIR HSI, acquiring data between 328 and 1115 nm. SPA and uninformative variable elimination (UVE) based on regression coefficients of PLS was used to select feature wavelengths while GLCM was used for textural extraction. RF and SIMCA were chosen as the classifiers. The importance of data fusion in terms of combining spectral and textural data to obtain higher classification rates between different groups was emphasized in the study [97]. The freshness of soaked and fresh prawn samples either in unfrozen or frozen states was determined using VIS-NIR HSI operating between 300 and 1100 nm. SPA was used to choose feature wavelengths while LS-SVM, adaptive boosting (AdaBoost) algorithm, and back-propagation neural networks (BP-NN) were used to build classification models using both full spectra and selected wavelengths. LS-SVM models utilizing full second derivative spectra have resulted in the highest correct classification ratio [98]. NIR-HSI acquiring data between 308 and 1105 nm was employed to discriminate fresh, cold-stored, and frozen-thawed grass carp fish fillets. Compared to other applied pre-processing methods such as multiplicative scatter correction, standard normal variate, and the second derivative, the highest correct classification ratio was obtained using first derivative spectra for building SIMCA, PLS-DA, LS-SVM, and PNN models. SPA was the method of choice to select feature wavelengths [99].

SW-NIR-HSI operating between 400 and 1000 nm was employed to analyze the shelf-life of vacuum-packed smoked salmon fillets. PLS-DA models were built with three different approaches, namely random pixel selection (without filter), spatial mean (basic filter), and pixel selection based on color (advance filter) [100]. Freeze–thaw history of vacuum-packed cod samples was analyzed using HSI operating at interactance acquisition mode between 430 and 1000 nm. Unlike previous studies, this work targeted to perform measurements on samples in the frozen state through the equilibration with −20 °C, and better classification results were obtained using the data acquired in the frozen state rather than the thawed state. Different procedures were followed for freezing, thawing, and frozen storage of the samples. Fast and slow freezing, as well as quick and slow thawing, was applied to different sample groups. Whole spectra, data between 450–600 nm, and data between 900–990 nm were utilized individually for PCA and k-NN models, while superior results were obtained when using entire spectral data [101].

Differentiation of fish species and determination of fish freshness was aimed at developing methods based on multimode HSI techniques. The efficiency of short-wavelength infrared (SWIR), VIS-NIR, fluorescence, and Raman HSI techniques was investigated within the context of the study. The higher sensitivity of fluorescence and Raman images to the tissue variations was revealed. Models utilizing full spectra, first ten PCs, or feature wavelengths were built using decision trees, discriminant analysis, naive Bayes classifiers, SVM, k-NN classifiers, and ensemble classifiers. Wavelength intervals were 419 to 1007 nm, 438 to 718 nm, and 842 to 2532 nm for VIS-NIR, fluorescence, and SWIR measurements, respectively. Raman spectra were collected from 103 to 2831 cm$^{-1}$. Feature wavelengths were selected using PCA and SFS. The highest classification accuracy was obtained by whole spectra VIS and NIR models for fish species differentiation and full spectra SWIR model for fish freshness determination [102]. Shrimp muscle samples from low-salinity freshwater and seawater farms were discriminated using NIR-HSI, acquiring images between 874 and 1734 nm. Potential of SPA, sequential forward selection (SFS), random frog (RF), and CARS were compared in terms of their ability to select feature wavelengths. Additionally, a deep selection process employing a correlation coefficient threshold between selected wavelengths was used to eliminate the selected wavelength that is still highly

correlated with the rest of the selected wavelengths. Classification models were built by utilizing either selected wavelengths or whole spectra based on PLS-DA, LS-SVM, or ELM methods. The best performance was obtained for the SFS-PLS-DA model. Using Pearson correlation analysis, a correlation was established between higher $^{13}C$ content of samples from high-salinity water and the corresponding hyperspectral data at 918–925 nm [103]. VIS-NIR HSI acquiring image between 393 to 1009 nm coupled with machine learning algorithms was employed to monitor the alterations in external appearances of live rainbow trout induced by two different diets containing either fish oil or plant oil during their cultivation. Performance of SVM models constructed with different preprocessed data was compared, and the best results were obtained through the SVM modeling of data treated by Savitzky-Golay smoothing and first derivative baseline removal. It is emphasized in the study that investigating the effects of different diets, nutrient deficiency, disease, age, and other environmental factors on fish skin properties may be evaluated as the subjects of future studies [104].

Feature wavelengths that have been used to discriminate fish species or determine fish freshness are summarized in Table 6. Lipids, carotenoids, and proteins are the molecules that are mostly responsible for the selection of determined wavelengths. When the results of the current literature on the use of HSI for food adulteration determination are evaluated with a bird's eye view, there is a need to say that there is huge diversity of both data analysis and variable selection methods. Despite the variety of data analysis methods that can be preferred for analyzing the information obtained using HSI, it seems that there is still a need for improvement in the assignment of characteristic signals so that this method can better define the analyzed sample. In this context, it should be taken into account that the new trends in the HSI method, which will be explained in the next section of the current article, may meet this need.

In the context of this review article, the application of hyperspectral imaging to different food product categories was discussed. It is clear from the literature that PCA and PLS-DA are still the two wide-spread chemometric methods applied to HSI data. The available literature has reached some maturity to compare the adequacy of chemometric methods in analyzing HSI data. In order to eliminate the different disadvantages of these methods, it is seen that a massive number of algorithms based on different complex mathematical foundations have been developed in recent years. The necessity of these developments is indisputably important. Since the major drawback of PCA modelling is the neglection of the spatial data, efforts have been focused on compensating for this drawback through the combination of PCA with different algorithms such as fixed size moving window-evolving factor analysis, deep learning algorithms, superpixelwise PCA approach, (SuperPCA), Gabor filtering, etc. Despite the numerous advantages of the incredibly diverse variety of algorithms reported by the authors in different studies, it currently does not seem sensible to compare the results and recommend the algorithm that will provide the greatest success in identifying a particular type of adulteration. In terms of building know-how, it is thought that it would be useful to test the performance of the developed data analysis technique over the quality of the obtained data acquired at different experimental environments and acquired from foods with different physical structures such as solid/liquid, granule/powder, fresh/frozen, packaged/unpackaged, etc. The only purpose of the studies should not be to determine the algorithm that gives the highest correlation and classification coefficient and to report the relevant results. Although the HSI technique seems to be suitable for integrating into industrial production conditions, considering its practical application, the know-how provided by the relevant literature on this subject does not seem sufficient in its current form. In order for HSI-based techniques to take place more often in industrial settings, the collection, storage, and processing of high-dimensional data remain an important problem. In addition, the effect of the conditions in which the data were collected on the success of the analysis needs to be investigated further.

**Table 6.** Feature wavelengths reported for fish and fishery products.

| Type of Adulteration | Spectral Range (nm) | Selected Wavelengths (nm) | Wavelength Selection Method | Assignment of the Wavelengths | Reference |
|---|---|---|---|---|---|
| Discrimination between fresh and frozen-thawed fish samples | 380–1030 | 729, 836, 970<br>928<br>512 and 620 | Savitzky–Golay 2nd derivative of absorbance data | Water content<br>Lipid and Protein content<br>Heme pigments | [95] |
| Discrimination between fresh, cold stored and frozen-thawed shelled shrimp samples | 328–1115 | 500<br>800<br>416, 435, 452, 478, 639, 689, 783, 813 | Uninformative variable elimination (UVE)-SPA | Water content<br>Astaxanthin content<br>No assignment. | [97] |
| Discrimination between fresh and soaked prawn samples in frozen and unfrozen states | 300–1100 | 420–460<br>530–580<br>950–1010<br>428, 504, 546, 556, 1000 | SPA of 1st derivative spectra | Astaxanthin content<br>Metmyoglobin content<br>Moisture content<br>Wavelengths discriminating unfrozen-fresh versus unfrozen-soaked samples | [98] |
| Discrimination between fresh, cold-stored and frozen-thawed grass carp fish fillets | 308–1105 | 560<br>970<br>446, 528, 541, 596, 660, 759, 970 | SPA of 1st derivative spectra | Astaxanthin and Canthaxanthin contents<br>Water content<br>Wavelengths discriminating fresh and stored samples | [99] |
| Discrimination between shrimp samples from freshwater and seawater farms | 874–1734 | 918965<br>1605, 1612, 1700<br>1656 | Deep selection process applied with SPA, CARS, Random Frog (RF) and sequential forward selection (SFS) | 3rd overtones of functional groups C-H/N-H/O-H stretching of organic components<br>2nd overtone of ester C=O vibration<br>1st overtone of C-H and its deformations of protein and glycogen<br>1st overtone of double bonds of vinyl groups (C=C) or aromatic rings of C-H stretching (Flavor difference) | [103] |
| Discrimination of live rainbow trout that are on different diets | 393–1009 | 450–750<br>900–1000 | Visual inspection | Lipid source influences the absorption and deposition of carotenoids | [104] |
| Differentiation of fish species and determination of fish freshness | VIS-NIR 419–1007 nm<br>SWIR 842–2532 nm<br>Fluorescence 718–84 2 nm<br>Raman 103–2831 cm$^{-1}$ | VIS-NIR 546, 560, 578<br>VIS-NIR 636<br>SWIR 984<br>SWIR 1208<br>Fluorescence ~470, 500, 530, 560, 590, 620, 650, 680, 700<br>Raman 487, 636, 734, 1097, 1311, 1451, 1651, 2305, 800–1000 | PCA | VIS-NIR Hemepigments<br>VIS-NIR Methemoglobin<br>SWIR water content<br>SWIR fat content<br>Fluorescence protein–protein interactions, Collagen structures<br>Raman unsaturated lipid composition | [102] |

PCA: principal component analysis, SPA: successive projection algorithm, CARS: competitive adaptive reweighted sampling, SWIR: short-wavelength infrared.

## 3. New Trends in HSI Based Methods

There are many research and review articles on the use of HSI-based techniques in food analysis. Some important points and new trends emphasized separately in these studies are brought together within the scope of the current article.

### 3.1. Hyperspectral Image Processing

The significant steps of an HSI operation are ordered as follows: data acquisition strategies, the hardware of HSI, hyperspectral image acquisition in different modes, image and spectral processing, classification and regression models, and visualization of chemical imaging [105]. Image processing is an essential step of HSI since it facilitates the removal of useless information and affects the performance of subsequent classification and prediction analyses as well as the quality of chemical imaging. Details of the methods have been described for hyperspectral image preprocessing composed of hyperspectral

image calibration, reduction of data size and image cleaning, spectral preprocessing techniques, and algorithms for spectral unmixing, and different ways of hyperspectral image post-processing were explained [106].

### 3.2. Selecting Feature Wavelengths to Design Multispectral Imaging Instruments

To design on-line multispectral sensors for specific purposes, the importance of selecting feature wavelengths out of high dimensional hyperspectral data has been emphasized in the literature. In a very detailed study, wavelength selection techniques were classified into three categories, which are filter methods, wrapper methods, and embedded methods. Principals and comparisons of these techniques can be found in the related study. Feature wavelengths for predicting food quality attributes that are determined using PLSR, SWR, SPA, UVE, spectrum derivative, and band ratio methods, artificial neural network (ANN) and its derivative methods such as causal index (CI) of the trained ANN model, feed-forward back-propagation ANN models, simulated annealing (SA), genetic algorithm (GA), CARS, receiver operating characteristic (ROC) analysis, branch and bound (BB) algorithm, minimum redundancy-maximum relevance (MRMR), and adaptive branch and bound algorithm (ABB) were summarized in the study [107].

### 3.3. Application of Regression Methods in HSI

Chemometric algorithms are mainly used to establish a reliable relationship between the quality attribute and the collected hyperspectral data. Details about the principles of linear and nonlinear quantitative regression algorithms and applications for meat quality detection have been explained. Multiple linear regression (MLR), principal component regression (PCR) and partial least squares regression (PLSR) are the most widely used examples of linear regression algorithms while SVM and ANN are the most commonly used nonlinear regression algorithms. Details of other methods such as LS-SVM, back-propagation neural network (BPNN) developed by the derivatization of these algorithms were also described [108]. Precautions that need to be followed in order to establish robust calibration models are listed below.

An essential issue for natural products is that bulk concentrations are considered for the construction of calibration range, while prediction is made based on pixel concentration. Using samples that are highly representative of the analyzed property is much more essential rather than using too much sample in the construction of the calibration model [109].

Targeting a specific adulterant rather than non-targeted food fraud testing, improper sampling, and non-reproducibility of measurements were defined as the three main obstacles that most of the studies failed to overcome. These points should be taken into consideration to establish a proper validation and employ the developed method in practice [110].

The immense importance of prediction data sets used to test the performance of calibration models was pointed out in the literature to avoid optimistic calibration results, which may be caused by the presence of features unrelated to the responses [60]. In addition to the use of an external test set, cross-validation must be performed to monitor the stability of the model, detect outlier samples, and get information about plausible sources of variation [109].

### 3.4. Snapshot Hyperspectral Imaging

To satisfy the need for low-cost and highly compact HSI cameras, single shot, or so-called snapshot hyperspectral sensors are developed using semiconductor process technology [111]. They allow imaging at video rate without the need to move the camera (detector) or sample (platform). In other words, using the snapshot technique, it is possible to obtain a hyperspectral image without scanning [112]. Their miniature size provides high portability. These features allow these sensors to get closer to the real-time applications, remote, and mobile HSI systems. The high speed of snapshot hyperspectral sensors

compared to conventional HSI systems constitutes promising potential for future industrial uses, while efforts are still needed to enhance the limited spatial resolution [72]. One of the recent results of these efforts is a hybrid camera setup in which a low spatial resolution HSI and a high spatial resolution conventional RGB image are combined with proper geometric alignment, and this approach is called HSI-super-resolution [113].

### 3.5. Alternative Imaging-Based Methods

The use of digital images captured by simple commercial devices to obtain chemical information is called digital image-based chemical analysis and has been employed to detect food adulteration [114]. It differs from multivariate image analysis since MIA uses only spectral features of the images [115]. It is reported that if computer vision systems (CVS) can be developed to obtain spectral information from foods, the long time required for data collection and processing in the HSI technique can be eliminated. It is stated that CVS offers promising potential for on-site and real-time food authentication [116]. On the other hand, maintenance of camera settings and adequate resolution of the photographic equipment to capture the analyzed characteristics are some of the essential parameters that need to be considered in terms of feasibility of visible imaging techniques to in situ and their real-time implementation [117].

### 3.6. Data Fusion

Zhou et al. have surveyed the effect of information fusion on the performance of food quality authentication studies. Although the superiority of fusion-based methods for food authentication was emphasized by the authors [118], it was not possible to generalize this positive effect of data fusion for methods coupled with HSI [119]. It is also pointed out that the cost–benefit ratio has to be considered in data fusion approaches [4].

## 4. Conclusions and Future Perspective

The HSI technique provides both spatial and spectral information as a result of one measurement. The high potential of HSI to fulfill the needs of industrial food control and sorting systems was ascertained almost a decade ago. However, high dimensional hyperspectral data with redundant information is still standing as the main challenge for this technique to be used for real-time monitoring in the food industry. Especially in the case of classification, the required number of samples for the training data set increases in tandem with the increasing dimensionality of the feature space. This fact about HSI is the main cause of the Hughes phenomenon and reported studies in the literature that employ either feature selection or feature extraction methods to overcome this problem. However, there is too much diversity and a lack of standardization in both preprocessing methods and data analysis algorithms used for both size reduction and visualization. Developments are still needed in hardware and artificial algorithms for faster data acquisition and more rapid data analyses. Besides, the complex nature of food products complicates the task of both food and data scientists. The presence of a possible relation between the algorithms developed to reduce the data size and the collected data (data acquisition system, food from which the data is collected, ambient conditions, etc.) should be questioned. In this way, applicable recommendations and know-how transfer will be provided, especially for industrial applications in the future.

The integration of HSI based systems to production lines is a hot topic; therefore, the efficiency of these techniques in the different steps of production lines (before/after packaging, etc.) should be studied in the future. Studies performed in plant or industrial environments rather than a laboratory may provide experience revealing the additional factors that should be given consideration. Additionally, studies discussing the significance of the difference between lab-scale and plant-scale performances of the developed method (from data acquisition to data analysis) are of great importance in terms of revealing the development areas of HSI-based techniques for industrial applications.

The use of Raman HSI in food analysis has come to a certain level, especially for powdered foods. However, there is still space for exhibiting its potential in the analysis of different food products. Enhancement of Raman signals through high laser power and long integration times are the first solutions that come to mind. Still, alternative solutions are required to mature this technique for its use in routine analyses of the food industry. Need for development in the database, hardware, image preprocessing, and processing algorithms is being felt for both HSFI and Raman HSI. Customizing the capabilities of fluorescence sensors may enhance their potential use for real-time monitoring. The combined use of hyperspectral fluorescence data with other measurement modes such as hyperspectral reflectance, etc., may also enhance the accuracy of the developed models.

**Author Contributions:** H.T.T. drafted the manuscript, designed the study, and interpreted the results. B.U. created the tables and interpreted the results. All authors have read and agreed to the published version of the manuscript.

**Funding:** This research received no external funding.

**Conflicts of Interest:** The authors declare no conflict of interest.

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
