# Peer review of "A Review of Recent Studies Employing Hyperspectral Imaging for the Determination of Food Adulteration"

_2673-7256, doi:10.3390/photochem1020008_

Round 1

Reviewer 1 Report

Manuscript ID: photochem-1260729 is a review article that focuses on the most  recent studies employed hyperspectral imaging  for the determination of food adulteration and authentication. The control of products identity is of great interest to avoid fraud and food scandals. The article is written in very good English and in general is well organized. It uses the most current literature in this topic and would be a very useful manual for new researchers in the field.

In this context, it addresses the limitations and future challenges in a very clear way. I have, however, some suggestions for authors to improve their study. These are the following:

  1.  The reference style in the whole manuscript is not in accordance with the guidelines of MDPI journals.
  2. The authors must include a Table or a Figure referring to the limitations and advantages of the chemometric techniques used in this topic. It would be very useful for the readers to have a fast scan of the applicability of these techniques.

Based on the above, I suggest a minor revision prior to the article publication.

Author Response

Dear Reviewer,

We appreciate your valuable comments.

  1. References were corrected in the whole manuscript.
  2. Since there are numerous review articles comparing the performance of chemometric methods to analyze HSI data, the present review article has been focused on the capabilities of different wavelength selection methods used in these chemometric methods. Below some of the review articles comparing the efficiency of different chemometric methods were listed.
  • Advanced Techniques for Hyperspectral Imaging in the Food Industry: Principles and Recent Applications, Annual Review of Food Science and Technology, 2019.
  • Hyperspectral and multispectral imaging for evaluating food safety and quality, Journal of Food Engineering, 2013.
  • Hyperspectral imaging as an emerging process analytical tool for food quality and safety control, Trends in Food Science & Technology, 2007.
  • Recent Developments in Hyperspectral Imaging for Assessment of Food Quality and Safety, Sensors, 2014.
  • Regression Algorithms in Hyperspectral Data Analysis for Meat Quality Detection and Evaluation. Comprehensive Reviews in Food Science and Food Safety, 2016.
  • Data Handling in Science and Technology, Chapter 9 - Hyperspectral Imaging and Chemometrics: A Perfect Combination for the Analysis of Food Structure, Composition, and Quality, 2013.

    A detailed discussion has been added to the manuscript, upon the comment of Reviewer 1. Details may be found below.

In the context of this review article, the application of hyperspectral imaging to different food product categories was discussed. It is clear from the literature that PCA and PLS-DA are still the two wide-spread chemometric methods applied to HSI data. The available literature has reached some maturity to compare the adequacy of chemometric methods in analyzing HSI data. In order to eliminate the different disadvantages of these methods, it is seen that a massive number of algorithms based on different complex mathematical foundations have been developed in recent years. The necessity of these developments is indisputably important. Since the major drawback of PCA modelling is the neglection of the spatial data, efforts have been focused on compensating this drawback through the combination of PCA with different algorithms such as fixed size moving window-evolving factor analysis, deep learning algorithms, superpixelwise PCA approach, (SuperPCA), Gabor filtering, etc.  Despite the numerous advantages of the incredibly diverse variety of algorithms reported by the authors in different studies, it currently seems not sensible to compare the results and recommend the algorithm that will provide the greatest success in identifying a particular type of adulteration. In terms of building know-how, it is thought that it would be useful to test the performance of the developed data analysis technique over the quality of the obtained data acquired at different experimental environments and acquired from foods in different physical structures such as solid/liquid, granule/powder, fresh/frozen, packaged/unpackaged, etc. The only purpose of the studies should not be to determine the algorithm that gives the highest correlation and classification coefficient and to report the relevant results. Although the HSI technique seems to be suitable for integrating into industrial production conditions considering its practical application, the know-how provided by the relevant literature on this subject does not seem sufficient in its current form. In order for HSI-based techniques to take more place in industrial applications, the collection, storage, and processing of high-dimensional data remain an important problem. In addition, the effect of the conditions in which the data were collected on the success of the analysis needs to be investigated further.

Reviewer 2 Report

The authors present in their manuscript a literature review concerning the applications of hyperspectral imaging to the detection of food adulteration with the focus on the selected eight food commodities reported in 2019 and 2020. These are 1) tea, coffee, and cocoa, 2) nuts and seeds, 3) herbs and spices, 4) honey and oil, 5) milk and milk products, 6) meat and meat products, 7) cereal and cereal products, and 7) fish and fishery product.

While the second paragraph is a relatively simple summary of published articles, the third one provides information regarding new trends in developing the HSI-based methods. What I personally miss in the discussion is the authors' personal opinion and comment or even a more critical approach concerning the selection of specific chemometric preprocessing and data modeling methods. With this respect, the authors failed. Unfortunately, the complexity of hyperspectral data and, in particular, their amount make the analysis very demanding and time-consuming. In fact, not always proposed in the literature modeling methods and discussed in published articles are feasible or sometimes even reasonable. Some of them require optimization of several input parameters (e.g., artificial neural networks), and their applications are somewhat limited to laboratory scale and not to real-time analysis. Moreover, the authors neglect a very relevant issue related to data processing and their storage. While most applications report successful stories, in my opinion, the “push-brum” hyperspectral systems and their efficient coupling with conveyor belts, computer(s), and expert systems are not yet straightforward. Therefore, the major focus should be put on increasing the speed of data transmission and processing to make the online analysis more efficient.

My major recommendation is to flavor the revised version of the manuscript with more personal opinions of the authors, and they should present a more critical approach. Of course, this suggestion is assuming that they have substantial experience in the field and chemometrics.

Specific comments:

  • The authors should avoid using the keywords words from the title.

Author Response

Dear Reviewer,

We appreciate your valuable comments.

  1. Keywords were revised based on your comments.
  2. Regarding the related comments of the reviewer, the manuscript has been enhanced with the discussion of the HSI limitations.

Details may be found below.

  • In the context of this review article, the application of hyperspectral imaging to different food product categories was discussed. It is clear from the literature that PCA and PLS-DA are still the two wide-spread chemometric methods applied to HSI data. The available literature has reached some maturity to compare the adequacy of chemometric methods in analyzing HSI data. In order to eliminate the different disadvantages of these methods, it is seen that a massive number of algorithms based on different complex mathematical foundations have been developed in recent years. The necessity of these developments is indisputably important. Since the major drawback of PCA modelling is the neglection of the spatial data, efforts have been focused on compensating this drawback through the combination of PCA with different algorithms such as fixed size moving window-evolving factor analysis, deep learning algorithms, superpixelwise PCA approach, (SuperPCA), Gabor filtering, etc.  Despite the numerous advantages of the incredibly diverse variety of algorithms reported by the authors in different studies, it currently seems not sensible to compare the results and recommend the algorithm that will provide the greatest success in identifying a particular type of adulteration. In terms of building know-how, it is thought that it would be useful to test the performance of the developed data analysis technique over the quality of the obtained data acquired at different experimental environments and acquired from foods in different physical structures such as solid/liquid, granule/powder, fresh/frozen, packaged/unpackaged, etc. The only purpose of the studies should not be to determine the algorithm that gives the highest correlation and classification coefficient and to report the relevant results. Although the HSI technique seems to be suitable for integrating into industrial production conditions considering its practical application, the know-how provided by the relevant literature on this subject does not seem sufficient in its current form. In order for HSI-based techniques to take more place in industrial applications, the collection, storage, and processing of high-dimensional data remain an important problem. In addition, the effect of the conditions in which the data were collected on the success of the analysis needs to be investigated further.

  • HSI technique provides both spatial and spectral information as a result of one measurement. The high potential of HSI to fulfill the needs of industrial food control and sorting systems was ascertained almost a decade ago. But high dimensional hyperspectral data with redundant information is still standing as the main challenge for this technique to be used for real-time monitoring in the food industry. Especially in the case of classification, the required number of samples for the training data set increases in correlation with the increasing dimensionality of the feature space. This fact about HSI is the main cause of the Huges phenomenon and reported studies in the literature to employ either feature selection or feature extraction methods to overcome this problem.  However, there are too much diverseness and a lack of standardization in both preprocessing methods and data analysis algorithms used for both size reduction and visualization.  Developments are still needed in hardware and artificial algorithms for faster data acquisition and more rapid data analyses. Besides, the complex nature of food products complicates the task of both food and data scientists. The presence of a possible relation between the algorithms developed to reduce the data size and the collected data ( data acquisition system, food from which the data is collected, ambient conditions, etc.) should be questioned. In this way, applicable recommendations and know-how transfer will be provided, especially for industrial applications in the future.
  • The integration of HSI based systems to production lines is a hot topic; therefore, the efficiency of these techniques in the different steps of production lines (before/after packaging, etc.) should be studied in the future. Studies performed in plant or industrial environments rather than laboratory may provide experience revealing the additional factors that should be given consideration. Additionally, studies discussing the significance of the difference between lab-scale and plant-scale performances of the developed method (from data acquisition to data analysis) are of great importance in terms of revealing the development areas of HSI-based techniques for industrial applications.

Round 2

Reviewer 2 Report

Dear Authors,

I am satisfied with the revision.

Kind regards!